# Grafted Sertoli Cells Exert Immunomodulatory Non-Immunosuppressive Effects in Preclinical Models of Infection and Cancer

**DOI:** 10.3390/cells13060544

**Published:** 2024-03-19

**Authors:** Sara Chiappalupi, Laura Salvadori, Monica Borghi, Francesca Mancuso, Marilena Pariano, Francesca Riuzzi, Giovanni Luca, Luigina Romani, Iva Arato, Guglielmo Sorci

**Affiliations:** 1Department of Medicine and Surgery, University of Perugia, 06132 Perugia, Italy; sara.chiappalupi@unipg.it (S.C.); monicaborghi@live.com (M.B.); francesca.mancuso@unipg.it (F.M.); marilena.pariano@unipg.it (M.P.); francesca.riuzzi@unipg.it (F.R.); giovanni.luca@unipg.it (G.L.); luigina.romani@unipg.it (L.R.); iva.arato@unipg.it (I.A.); 2Interuniversity Institute of Myology (IIM), 06132 Perugia, Italy; laurasalvadori1988@gmail.com; 3Consorzio Interuniversitario Biotecnologie (CIB), 34127 Trieste, Italy; 4Department of Translational Medicine, University of Piemonte Orientale, 28100 Novara, Italy; 5Centro Biotecnologico Internazionale di Ricerca Traslazionale ad indirizzo Endocrino, Metabolico ed Embrio-Riproduttivo (CIRTEMER), 06132 Perugia, Italy; 6Centro Universitario di Ricerca sulla Genomica Funzionale (CURGeF), 06132 Perugia, Italy

**Keywords:** Sertoli cells, immunomodulation, fungal infection, tumor growth, metastasis

## Abstract

The Sertoli cells (SeCs) of the seminiferous tubules secrete a multitude of immunoregulatory and trophic factors to provide immune protection and assist in the orderly development of germ cells. Grafts of naked or encapsulated SeCs have been proved to represent an interesting therapeutic option in a plethora of experimental models of diseases. However, whether SeCs have immunosuppressive or immunomodulatory effects, which is imperative for their clinical translatability, has not been demonstrated. We directly assessed the immunopotential of intraperitoneally grafted microencapsulated porcine SeCs (MC-SeCs) in murine models of fungal infection (*Aspergillus fumigatus* or *Candida albicans*) or cancer (Lewis lung carcinoma/LLC or B16 melanoma cells). We found that MC-SeCs (i) provide antifungal resistance with minimum inflammatory pathology through the activation of the tolerogenic aryl hydrocarbon receptor/indoleamine 2,3-dioxygenase pathway; (ii) do not affect tumor growth in vivo; and (iii) reduce the LLC cell metastatic cancer spread associated with restricted *Vegfr2* expression in primary tumors. Our results point to the fine immunoregulation of SeCs in the relative absence of overt immunosuppression in both infection and cancer conditions, providing additional support for the potential therapeutic use of SeC grafts in human patients.

## 1. Introduction

Sertoli cells (SeCs) are the major component of the seminiferous tubules, where they provide the trophic factors required for the orderly development of maturing germ cells (the “nurse cell” role) and create a physical barrier (the blood–testis barrier; BTB) made of tight junctions between adjacent SeCs separating the germ cells from the blood supply [1,2]. The BTB protects developing germ cells (i.e., spermatocytes, spermatids, and spermatozoa) since they emerge at puberty when the immune system has already formed and express newly synthesized markers on their surface that could be recognized as foreign by the immune system itself. Besides forming the BTB, SeCs secrete several immunoregulatory factors that contribute to this immune protection. Indeed, SeCs secrete (i) factors that suppress interleukin (IL)-2 production, thus blocking the clonal expansion of activated lymphocytes [3,4]; (ii) transforming growth factor-β, which induces the differentiation of infiltrating cells into a Th2 (protective) over Th1 (destructive) response [5]; (iii) soluble JAGGED1, which contributes to the de novo generation of Tregs through the activation of the Notch pathway [6]; and (iv) other factors which contribute to restraining the immune system response such as activin, clusterin, macrophage inhibitory factor, and serpins [7]. Moreover, SeCs express the Fas ligand on their surface, which can downregulate the immune system by inducing apoptosis in Fas-bearing T cells [8], and express membrane-bound and soluble complement cascade inhibitors that protect germ cells against complement-mediated cell lysis [9]. SeCs constitutively express low levels of major histocompatibility complex I but not II antigens [10] and produce indoleamine 2,3-dioxygenase (IDO), an enzyme that catalyzes the oxidative cleavage of L-tryptophan to produce kynurenines, thus providing immune tolerance involving regulatory T cells [7,11]. The crucial role exerted by SeCs in the testis is highlighted by conditions in which a breakdown of testicular immune privilege occurs, resulting in testicular autoimmunity [12].

The peculiar immunological properties of SeCs have prompted researchers to investigate their use to protect allogeneic or xenogeneic grafts from immune destruction, thus prolonging the survival of transplants and avoiding their rejection, or to create a trophic and anti-inflammatory environment in multiple preclinical models of diseases [13,14]. In particular, microencapsulated SeCs (MC-SeCs) have been successfully grafted, in the absence of pharmacological immunosuppression, into the peritoneal cavity of several pre-clinical models of chronic diseases, including type 1 and type 2 diabetes, Huntington’s disease, Laron syndrome, and Duchenne muscular dystrophy [13,14,15]. The data obtained using these experimental models suggest that grafted SeCs exert an immunomodulatory rather than immunosuppressive role, which is an important issue in translational medicine and clinical perspectives. However, a clear demonstration of this important aspect of SeCs has not been provided so far [12]. 

Here, we have addressed this issue by directly assessing the immunopotential of microencapsulated porcine SeCs (MC-SeCs) in murine models of fungal infections and cancer. We found that MC-SeCs exhibited a fine-grained immunoregulatory activity in both infection and cancer by providing antifungal resistance with minimum pathology and reducing metastatic cancer spread. These results point to the potent immunomodulatory activity of SeCs in the relative absence of overt immunosuppression.

## 2. Materials and Methods

### 2.1. Cell Cultures

Murine Lewis lung carcinoma (LLC) and melanoma (B16-F10) cells purchased from ATCC were cultured in high-glucose DMEM containing P/S and supplemented with 10% fetal bovine serum. The LLC and B16 cells were maintained in a humidified atmosphere containing 5% CO_2_ at 37 °C.

### 2.2. Animal Models of Tumor Growth and Fungal Infection

C57BL/6 mice purchased from Charles River Laboratories (Calco, Italy) were raised on a 12 h light/day cycle and a standard diet. Six- to eight-week-old male and female mice weighing 20–25 g were used in all the experiments. The mice were injected i.p. with MC-SeCs (1.0 × 10^6^ SeCs/g body weight) or the equivalent amount of empty microcapsules (E-MCs) using a sterile 16-gauge catheter under general anesthesia [15] one week before fungal infection or tumor cell injection. For *Aspergillus fumigatus* infection, mice previously injected with MC-SeCs or E-MCs were anesthetized in a plastic cage via inhalation of 3% isoflurane (Forane; Abbott Laboratories, North Chicago, IL, USA) in oxygen before the intranasal instillation of 4.0 × 10^7^ resting conidia or intratracheal instillation of 5.0 × 10^7^ resting conidia of *A. fumigatus* (Af293 strain) in 20 µL of saline. Gastrointestinal *Candida albicans* infection was performed via gavage of 1.0 × 10^8^ *C. albicans* yeasts in 200 µL of saline for two consecutive days. At the indicated times, the animals were sacrificed, and their lungs, bronchoalveolar lavage fluid (BALF), or colons were collected. For tumor growth analysis, mice previously injected with MC-SeCs or E-MCs were subcutaneously (s.c.) injected in the right flank with LLC (1.0 × 10^6^ cells/mouse) or B16 (1.0 × 10^5^ cells/mouse) cells. The control mice were injected with vehicle. Tumor measurements were taken using a digital caliper (Exacta Optech Labcenter, San Prospero, Italy), and the tumor volumes were calculated using the formula (length × width^2^)/2. Twenty-five days after tumor cell injection, the animals were sacrificed, and their tumor masses and lungs were excised and weighed. The numbers and areas of lung metastases were evaluated after hematoxylin/eosin staining. 

### 2.3. SeC Purification and Characterization

The SeCs were isolated from pre-pubertal Large White pig testes and encapsulated in alginate microcapsules, as reported [16]. Briefly, after their removal under anesthesia, the testes were minced and digested with collagenase P (2 mg/mL; Roche Diagnostics, Monza, Italy) for 20 min at 37 °C, followed by digestion with trypsin and DNase I (Sigma-Aldrich, St. Louis, MO, USA) for 10 min at 37 °C. Residual Leydig and peritubular cells were eliminated according to resuspension of the pellets in glycine. The resulting SeCs were cultured in HAM’s F-12 (Euroclone, Milan, Italy) supplemented with retinoic acid (0.166 nM; Sigma-Aldrich) and insulin–transferrin–selenium (ITS, Becton Dickinson, Franklin Lakes, NJ, USA) (1:100) in a humidified atmosphere containing 5% CO_2_ at 37 °C. After 3 days of culture, residual germ cells were eliminated via incubation with 10 mM tris-hydroxymethyl-aminomethane hydrochloride (Tris-HCl; Sigma-Aldrich) buffer. The purified SeCs were characterized by evaluating the anti-Müllerian hormone, insulin-like 3, α-smooth muscle actin, and protein gene product 9.5 expression. The SeC functionality was evaluated by measuring the α-aromatase activity. 

### 2.4. Preparation of the MC-SeCs

To envelop the SeCs in barium alginate microcapsules (MC-SeCs), freshly prepared SeCs were collected after incubation and suspended in 1.6% aqueous solution of sodium alginate (1.6–2.0% *w*/*v*, viscosity 120–150 mPas; Keltone^®^ LVCR; FMC BioPolymer, Erie, PA, USA) with an endotoxin content lower than 0.5 EU/g. The cell suspension was continuously mixed to prevent cell aggregation and obtain a homogeneous distribution of SeCs within the alginate solution, and the suspension was aspirated using a peristaltic pump (flow rate, 14 mL/min) and extruded through a microdroplet generator (air flow rate, 1.5–2 L/min) under sterile conditions. The SeCs’ viability inside the microcapsules was evaluated after ethidium bromide/fluorescein diacetate double staining, and the microcapsule morphology and size distribution were assessed before injection, as described [15]. Only preparations with at least 95% viable SeCs were used in all the experiments.

### 2.5. Histology and Immunohistochemistry

For histology, the organs were removed and immediately fixed in 10% neutral buffered formalin (Bio Optica, Milan, Italy) for 24 h. The fixed organs were dehydrated, embedded in paraffin, sectioned into 3–4 μm sections, and stained with hematoxylin–eosin, periodic acid–Schiff, or Grocott reagent (Bio Optica). 

For immunohistochemistry, sections of the formalin-fixed paraffin-embedded lungs were deparaffinized with xylene and rehydrated in a graded ethanol series. Antigen retrieval was performed by boiling them for 1.5 h in 10 mM citric acid buffer (pH 6.0), and depletion of endogenous peroxidase was accomplished through treatment with 3% H_2_O_2_. The sections were incubated overnight in a humid chamber at 4 °C with the anti-Ki-67 mouse specific monoclonal antibody (D3B5 clone, IHC-formulated; Cell Signaling Technology, Danvers, MA, USA) diluted 1:400 in blocking buffer (BB; Tris-buffered saline containing 0.1% Tween-20 and 10% horse serum). The day after, the sections were incubated with an anti-rabbit biotinylated antibody (Vector Laboratories, Newark, CA, USA; 1:500 dilution) for 1 h in BB. The sections were incubated for 45 min with VECTASTAIN ABC-HRP kit reagents (Vector Laboratories, Newark, CA, USA), followed by incubation with 0.01% 3-diaminobenzidine tetrahydrochloride (DAB) and 0.006% H_2_O_2_ in 50 mM Tris-HCl (pH 7.4). The nuclei were counterstained with hematoxylin. Dehydrated sections were mounted using Eukitt medium (Electron Microscopy Sciences, Hatfield, PA, USA) and photographed using an Olympus BX51 microscope equipped with a digital camera.

### 2.6. Morphological Quantification

For histological evaluation of lung aspergillosis, the lung sections were examined in a blinded manner and scored based on the numbers and extent of the inflammatory foci and damage to the parenchyma architecture. The scores were assigned as follows: 0, normal; 1, minor; 2, mild; 3, severe. The extent of colon candidiasis was expressed as the inflammatory cell infiltration area. The lung metastasis numbers and areas were evaluated on the hematoxylin/eosin-stained sections at 100 μm intervals for the entire organ. The inflammatory cell infiltration areas and lung metastasis areas were measured using the freely downloadable ImageJ software version 1.51j8 (https://imagej.nih.gov/ij/).

### 2.7. Bronchoalveolar Lavage Fluid (BALF) Morphometry

The lungs were filled thoroughly with 1.0 mL aliquots of pyrogen-free saline using a 22-gauge bead-tipped feeding needle introduced into the trachea. For differential BALF cell counts, cytospin preparations were stained using May-Grünwald/Giemsa reagents (Sigma-Aldrich, St. Louis, MO, USA). At least 10 fields were counted, and the percentages of the lymphocytes, monocytes, eosinophils, and polymorphonuclear cells were calculated. Photographs were taken using a high-resolution Olympus BX51 microscope with a 100× objective.

### 2.8. Real-Time PCR

The lungs, colons, and tumor masses were lysed, and total RNA was extracted using the RNeasy Mini Kit (QIAGEN, Milan, Italy) and reverse-transcribed using the PrimeScript^TM^ Reagent Kit with the gDNA Eraser (Perfect Real Time) kit (Takara Bio Europe, Saint-Germain-en-Laye, France), according to the manufacturer’s instructions. For the lungs and colons, real-time PCR analyses were performed using the iCycler iQ detection system (Bio-Rad, Hercules, CA, USA) and iTaq™ Universal SYBR^®^ Green Supermix (Bio-Rad, Hercules, CA, USA). The amplification efficiencies were validated and normalized against *Actb* (β-actin). The thermal profile for the SYBR Green real-time PCR was 95 °C for 3 min, followed by 40 cycles of denaturation at 95 °C for 30 s and annealing/extension at 60 °C for 30 s. For the tumor masses, real-time PCR analyses were performed with Stratagene Mx3000P (Agilent Technologies, Santa Clara, CA, USA) using the HOT FIREPol EvaGreen qPCR Mix Plus (ROX) ready-to-use solution (Solis BioDyne, Tartu, Estonia). The amplification efficiencies were validated and normalized against *Gapdh*. The thermal profile for EvaGreen was 95 °C for 12 min, followed by 40 cycles of denaturation for 15 s at 95 °C, annealing at 60 °C for 20 s, and extension at 72 °C for 20 s. Each data point was examined for integrity through analysis of the amplification plot. The relative quantity (ΔΔCt method) was evaluated based on the dRn (i.e., baseline subtracted fluorescence reading normalized to the reference dye) as calculated using the dedicated Stratagene MxPro software version 4.10d. The primers used in the real-time PCR analyses are reported in Appendix A.

### 2.9. Statistical Analysis

Student’s *t*-test and one-way or two-way analysis of variance (ANOVA) with the Bonferroni post hoc test was used to determine the statistical significance, defined as *p* < 0.05. The data are shown as pooled results (mean ± SD or mean ± SEM) or representative images from two or three experiments. The histological scores were compared using the Kruskal–Wallis test. The in vivo groups consisted of 3 to 6 mice/group. The mice were allocated into each group according to simple randomization. No including/excluding criteria were set for selecting animals during the experiments. No animals were excluded from the analysis. Quantifications were performed by three independent operators blinded to treatments. GraphPad Prism software V.6.01 was used for the analysis.

### 2.10. Ethics Approval

This study was performed in line with the principles of the Declaration of Helsinki. Approval was granted by the Ethics Committee of the University of Perugia and the Italian Ministry of Health (468/2017-PR, 6 June 2017).

## 3. Results

### 3.1. Grafted MC-SeCs Restrain Immunopathology in Infection

To assess the immunomodulatory function of the SeCs, we resorted to experimental models of fungal infections, in which the immune response and its modulation play an important role [17]. *Aspergillus fumigatus* and *Candida albicans* are the natural commensals that most commonly cause invasive fungal infections in immunocompromised individuals, often culminating in patients’ death [18]. Thus, we injected the C57BL/6 mice with microencapsulated porcine SeCs (MC-SeCs) (equivalent amount, 1.0 × 10^6^ SeCs/g body weight) or the same amount of empty microcapsules (E-MCs) intraperitoneally (i.p.) and then infected the animals with either *A. fumigatus* intranasally or *C. albicans* intragastrically (Figure 1A and Figure 2A). The mice were monitored for fungal growth and histopathology in the target organs (the lungs in aspergillosis and the colon in candidiasis), as were the parameters of immune resistance and tolerance, i.e., the patterns of inflammatory/anti-inflammatory cytokine production and the combined activity of the aryl hydrocarbon receptor (AhR)/indoleamine 2,3-dioxygenase (IDO) pathway, known to contribute to immune tolerance in infection [17]. We found that the administration of MC-SeCs protected against aspergillosis, as revealed by the reduced inflammatory pathology in the lungs (Figure 1B,C), associated with the reduced fungal growth (Figure 1B,D). Of great interest, MC-SeCs promoted the recruitment of mononuclear cells rather than inflammatory neutrophils, as indicated by the BALF morphometry and cell counts (Figure 1B,E), the reduced expression of the neutrophil-recruiting chemokine *Cxcl1* and neutrophil markers (*S100a8* and *S100a9*), and the increased expression of the monocyte-recruiting *Ccr2* chemokine (Figure 1F). 

On assessing the patterns of inflammatory/anti-inflammatory cytokine gene expression, we found that the expression levels of IL-6 (*Il6*), IL-1β (*Il1b*), and IL-17A (*Il17a*) were significantly reduced by the MC-SeCs, whereas the expression of anti-inflammatory mediators, such as *Ido1*, *Il10*, and *Il1rn*, were significantly increased during the post-infection period (Figure 1G). Administration of the MC-SeCs also prevented immunopathology in mice severely infected with *A. fumigatus*, such as in mice injected intratracheally with the fungus (Appendix A). These results suggest that MC-SeCs may promote antifungal effector activity in the lungs with minimum inflammation and pathology. 

As in aspergillosis, the administration of MC-SeCs also regulated inflammation and prevented immunopathology in gastrointestinal candidiasis (Figure 2A), as revealed by the reduced inflammatory pathology and restored tissue architecture in the colon. Indeed, we found strongly reduced inflammation and an almost normal colon morphology at dpi 10 in the MC-SeC-grafted mice (average inflammatory areas of 0.096 × 10^3^ μm^2^, 5.89 × 10^3^ μm^2^, and 4.92 × 10^3^ μm^2^ in MC-SeC-injected, E-MC-injected, and control mice, respectively) (Figure 2B). In accordance, the colons of the mice injected with MC-SeCs showed a reduced expression of inflammatory mediators (Figure 2C) and an increased expression of the anti-inflammatory *Ido1*, *Il10*, and *Il1rn* cytokines (Figure 2D). 

The apparent ability of MC-SeCs to restore the tissue architecture in both target organs prompted us to assess whether the AhR-dependent pathway, known to promote mucosal barrier function and tissue homeostasis [19], was involved. For this purpose, we evaluated the expression of genes downstream of AhR, such as cytochrome P450 family 1 subfamily A member 1 (*Cyp1a1*) and subfamily B member 1 (*Cyp1b1*) and *Il22* in the lungs and colons of mice with aspergillosis and candidiasis, respectively. We found that the MC-SeCs but not the E-MCs promoted the expression of these AhR-dependent genes in colon fungal infection compared to the controls (Figure 2E). Similarly, *Cyp1b1* and *Il22* were upregulated by the MC-SeCs in the lung infection model (Figure 1H), suggesting the involvement of the AhR-dependent pathway also in this case. Altogether, these results pointed to the regulatory, non-immunosuppressive activity of the grafted MC-SeCs in inflammation and tissue repair in infection.

### 3.2. Grafted MC-SeCs Reduce Metastatic Cancer Spread

Because immunosuppression leads to the blockade of immune surveillance and increases the risk of developing tumors [20], we investigated the effects of SeCs in two widely used experimental models of cancer, i.e., C57BL/6 mice injected subcutaneously (s.c.) with Lewis lung carcinoma (LLC) or B16 melanoma cells [21,22]. In both cases, MC-SeCs (equivalent amount, 1.0 × 10^6^ SeCs/g body weight) or the same amount of E-MCs were injected i.p. 7 days before the injection of cancer cells, and the tumor growth was followed for an additional 25 days (Figure 3A). The presence of the MC-SeCs did not affect the cancer cell growth, as the tumor volumes measured over time (Figure 3B,D) and the tumor weights after excision at the time of sacrifice (Figure 3C,E) were similar in both the MC-SeC- and E-MC-treated mice. No major differences were found either between the tumor masses excised from the mice previously injected with MC-SeCs and those from the E-MC-injected mice in terms of their overall morphology and the extent of the necrosis areas (Appendix A). 

Interestingly, the LLC-bearing mice previously injected with MC-SeCs showed a remarkable reduction in lung metastasis formation, with lower numbers of metastases (2.4 ± 0.5 vs. 5.6 ± 0.7 lung metastases/mouse in MC-SeC- and E-MC-injected mice, respectively), which were characterized by strongly reduced sizes (0.1 ± 0.02 vs. 1.0 ± 0.2 mm^2^ in MC-SeC- or E-MC-injected mice, respectively) (Figure 4A–C). In line with the notion that B16 cells do not give rise to lung metastases when injected s.c. [23], we did not find any metastasis in the lungs of the B16-bearing mice.

The different sizes found for the LLC lung metastases were not the result of different proliferation extents since similar expressions of the proliferation marker Ki67 were found in the metastases of the MC-SeC- and E-MC-injected mice (Figure 4A; lower panels). Thus, we analyzed the LLC primary tumors for the expression of the transcription factors involved in epithelial–mesenchymal transition (i.e., *Snai1*, *Twist1*, and *Zeb1*) [24] and the matrix metalloproteases *Mmp2* and *Mmp9*, the increased expressions of which confer epithelial cells the ability to degrade extracellular matrix components and migrate [25]. Moreover, we evaluated the expression of the angiogenesis markers *Vegfa*, *Vegfb*, and *Vegfc* and their receptor vascular endothelial growth factor (VEGF) receptor 2 (*Vegfr2*; also known as kinase insert domain receptor, *Kdr*), which also contributes to cancer development and metastasis [26]. No statistically significant differences in the expression of *Snai1*, *Twist1*, *Zeb1*, *Mmp2*, *Mmp9*, *Vegfa*, *Vegfb*, or *Vegfc* were found between the MC-SeC- and E-MC-injected mice (Appendix A); however, an overall strong reduction in *Vegfr2* expression was found in the LLC masses developed in the MC-SeC- in comparison with the E-MC-injected mice (Figure 4D). Thus, the presence of grafted SeCs in the LLC-bearing animals, while not affecting the primary cancer cell growth, restrained the cancer cell spread and lung colonization, which may be linked to the reduced expression of *Vegfr2* in the primary tumors.

## 4. Discussion

The peculiar secretory properties of SeCs have prompted their use in a variety of experimental models of diseases, in which SeCs have been grafted naked or encapsulated to create a tolerogenic environment. However, the therapeutic potential of SeCs could be hampered by potential side effects linked to immunosuppression, including toxicity and an increased risk of developing infections and cancer [27,28]. In infection, although SeCs express a variety of pattern recognition receptors through which they may recognize pathogens and activate antimicrobial defenses [29,30], whether SeCs properly regulate the antimicrobial immune defense in vivo has not been demonstrated so far. We found here that SeCs provide resistance to fungal infections with minimum pathology, i.e., SeCs properly regulate antimicrobial resistance in vivo. This likely occurs through the recruitment of monocytes rather than inflammatory neutrophils and the activation of the AhR/IDO pathway, known to provide immune tolerance through a combination of physical and immunological barriers in the gut and lungs [17,31]. Considering that SeCs have phagocytic capabilities, being able to engulf pathogens without the elicitation of a concomitant inflammatory response [32,33], the results of our study further add to the multitasking activity SeCs may have in infections. However, the use of SeCs sequestered into alginate microcapsules and spatially confined to the peritoneal cavity of the host [13,16] leads us to conclude that secreted molecules with antimicrobial activity [14] rather than the direct activity of SeCs play a role in infection settings.

In cancer, the link between immunosuppression and cancer development is well recognized. In transplant recipients, the occurrence of cancers (especially non-Hodgkin lymphoma and cancers of the lungs, kidneys, and liver) is more common than in the general population [20,34]. In addition, the SeC secretome is characterized by the presence of trophic factors, including insulin-like growth factor-1, whose signaling could potentially contribute to tumor cell motility and promote cell invasion and metastasis [35]. However, in both LLC and B16 tumor-bearing mice, we found that grafted MC-SeCs did not affect the tumor growth over time, whereas reduced lung metastasis formation was observed in the LLC-bearing mice, suggesting that SeC-released factors may modulate the invasive properties of primary tumor cells. This is likely due to the reduced expression of the tyrosine kinase receptor *Vegfr2* observed in the LLC primary tumors of the MC-SeC-injected mice. Interestingly, the activation of VEGFR2 by VEGFs has been linked to a more aggressive tumor phenotype [26,36]. Whether SeC-derived factors also affect cancer cell lung colonization remains to be investigated. However, grafted SeCs seem not to affect metastatic cell proliferation since we found a similar Ki67 expression extent in the lung metastases of the mice injected with MC-SeCs and empty microcapsules.

The factors responsible for SeCs’ immunological properties are not completely known, although our knowledge of the immuneprivileged status of the testis dates back to 1767, when the father of scientific surgery John Hunter observed that rooster testes transplanted into the abdominal cavity of a hen maintained a “perfectly normal structure” [37]. Later, it became clear that this immuneprivileged status was dependent on the presence of SeCs, which, in addition to creating the BTB, secrete numerous soluble factors that modulate immune system responses [1,2]. For the same reasons, and thanks to the use of highly biocompatible clinical-grade alginate, MC-SeCs are resultantly invisible to the host immune system [13]. The emerging idea is that the effects exerted by SeCs are the result of a cocktail of factors (i.e., maturation factors, hormones, growth factors, cytokines, and immunomodulatory factors) whose exact formulation is difficult to establish and whose global effect is not just the simple sum of each factor’s activity [7]. Moreover, SeC activity may be influenced by the biological status of the cells and environmental conditions (e.g., inflammatory vs. physiological conditions). By analyzing the gene expression profile, Doyle and coworkers [38] identified three functions in the primary SeCs potentially important to establishing SeCs’ immune privilege: (i) the suppression of inflammation by specific cytokines and prostanoids; (ii) the reduction in leukocyte migration linked to cell junctions and actin polymerization; and (iii) the inhibition of complement activation and subsequent cell lysis. This suggests a scenario in which SeCs can modulate at the same time multiple aspects of the immune system responses. The data presented here clearly show that the cocktail of factors secreted by the SeCs exerts an overall immunomodulatory rather than an immunosuppressive effect.

The concept of SeCs as an immunomodulatory cell type is in line with the observation that no complications emerged in long-term analyses in animal models and in long-term clinical follow-up in patients after SeC grafts. Indeed, no overt side effects were observed in mice injected with MC-SeCs in long-term analyses and in type 2 diabetes non-human primates injected with MC-SeCs at a 6-month follow-up [7,15]. In support of the absence of SeC-related carcinogenetic effects, we observed that *mdx* mice, an animal model of Duchenne muscular dystrophy that spontaneously develops rhabdomyosarcomas over time [39], did not show increased tumor appearance even one year after MC-SeC injection, in comparison with E-MC-injected or untreated animals (unpublished data). Importantly, the group of Valdés-González transplanted several type 1 diabetic patients with porcine SeCs together with neonatal porcine pancreatic islets in subcutaneous devices; no complications or porcine endogenous retrovirus infection were detected in patients up to 7 years post-transplant, with porcine insulin detected in the patients’ sera following glucose stimulation and the presence of insulin-positive cells demonstrated 3 years post-transplant [40,41]. 

Our results further support the use of SeCs in human patients, which would represent a challenge in the clinical therapy since a plethora of chronic inflammatory diseases and autoimmune diseases could benefit from SeC-based therapeutic approaches.

## Figures and Tables

**Figure 1 cells-13-00544-f001:**
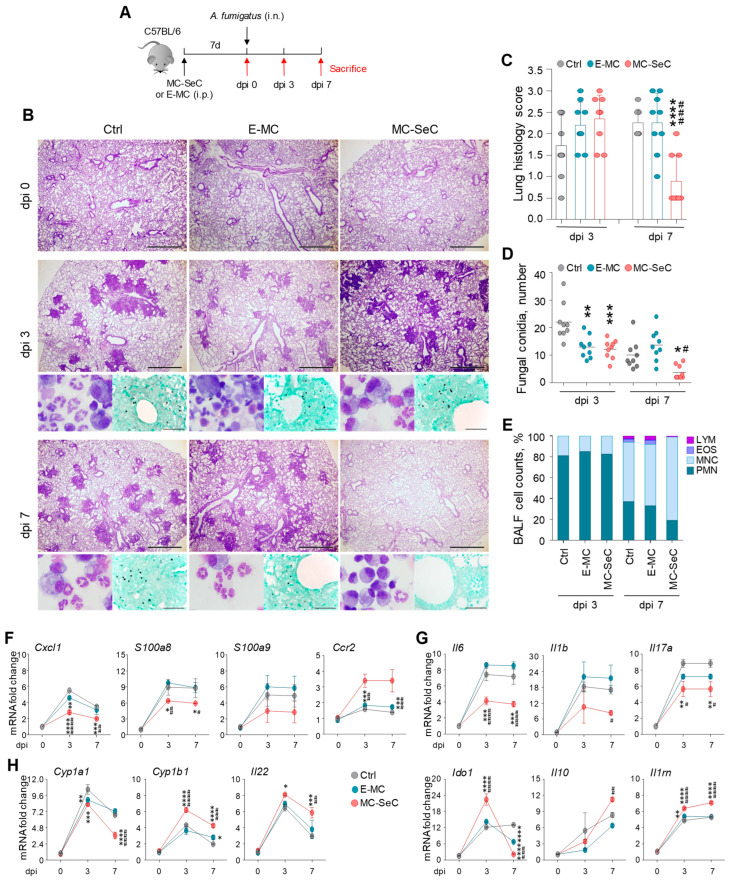
Grafted SeCs protect against *Aspergillus fumigatus* pulmonary infection. (**A**–**H**) C57Bl/6 mice previously injected intraperitoneally with microencapsulated porcine SeCs (MC-SeCs) (equivalent amount, 1.0 × 10^6^ SeCs/g body weight) or the same amount of empty microcapsules (E-MCs) were infected intranasally with *A. fumigatus* conidia and assessed at 0, 3, and 7 days post-infection (dpi) (**A**). (**B**) Periodic acid–Schiff staining (main panels) and Grocott staining (lower-right panels) in paraffin-embedded sections of lungs and BALF morphometry (lower-left panels). (**C**) Reported are the average pathology scores obtained for lungs based on histology. (**D**) The number of fungal conidia were counted after Grocott staining. (**E**) Differential cell counts in BALF are reported as percentages of lymphocytes (LYM), monocytes (MNC), eosinophils (EOS), and polymorphonuclear cells (PMN). (**F**–**H**) Neutrophil- and monocyte-recruiting chemokines (**F**), inflammatory and anti-inflammatory cytokines (**G**), and *Cyp1a1*, *Cyp1b1*, and *Il22* (**H**) gene expressions were evaluated using real-time PCR on total lung homogenates. Scale bars (**B**), 1 mm. Original magnification (**B**, lower-left panels), 100×. Shown are the results of pooled homogenates (n = 3, mean ± SD) or representative images from three independent experiments. * *p* < 0.05, ** *p* < 0.01, *** *p* < 0.001, **** *p* < 0.0001 vs. internal control (Ctrl) mice; ^#^
*p* < 0.05, ^##^
*p* < 0.01, ^###^
*p* < 0.001, ^####^
*p* < 0.0001, MC-SeC- vs. E-MC-injected mice. One- or two-way ANOVA, Bonferroni post hoc test.

**Figure 2 cells-13-00544-f002:**
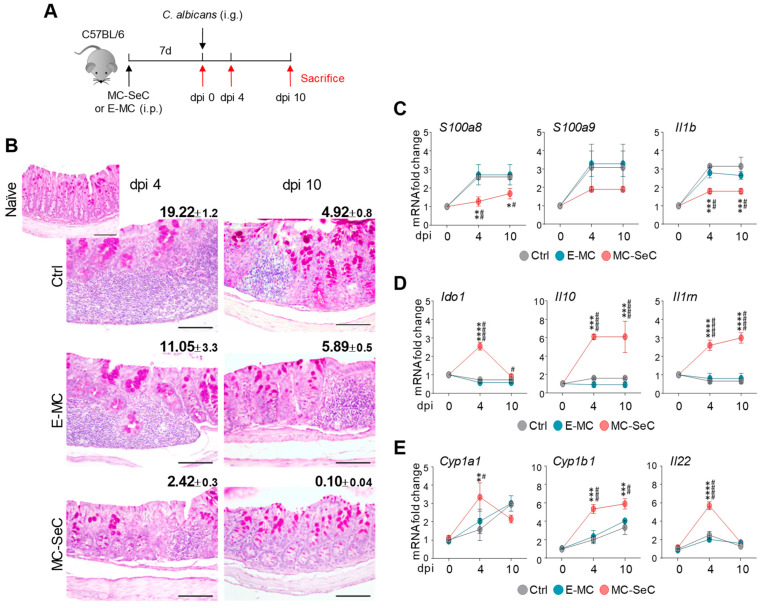
Grafted SeCs protect against *Candida albicans* gastrointestinal infection. (**A**–**E**) C57Bl/6 mice previously injected intraperitoneally with microencapsulated porcine SeCs (MC-SeCs) (equivalent amount, 1.0 × 10^6^ SeCs/g body weight) or the same amount of empty microcapsules (E-MCs) were infected with *Candida albicans* via gavage and evaluated at 4 and 10 days post-infection (dpi) (**A**). Periodic acid–Schiff staining of colons at different dpi (**B**). Reported are the average inflammatory cell infiltration areas (×10^−3^ μm^2^) ±SEM. Neutrophil markers and inflammatory cytokines (**C**), anti-inflammatory cytokines (**D**), and *Cyp1a1*, *Cyp1b1*, and *Il22* (**E**) gene expressions in colons were evaluated using real-time PCR. Shown are the results of pooled homogenates (n = 3, mean ± SD) and representative images from three independent experiments. * *p* < 0.05, ** *p* < 0.01, *** *p* < 0.001, **** *p* < 0.0001 vs. internal control (Ctrl) mice; ^#^
*p* < 0.05, ^##^
*p* < 0.01, ^###^
*p* < 0.001, ^####^
*p* < 0.0001, MC-SeC- vs. E-MC-injected mice. One- or two-way ANOVA, Bonferroni post hoc test. Scale bars (**B**), 100 μm.

**Figure 3 cells-13-00544-f003:**
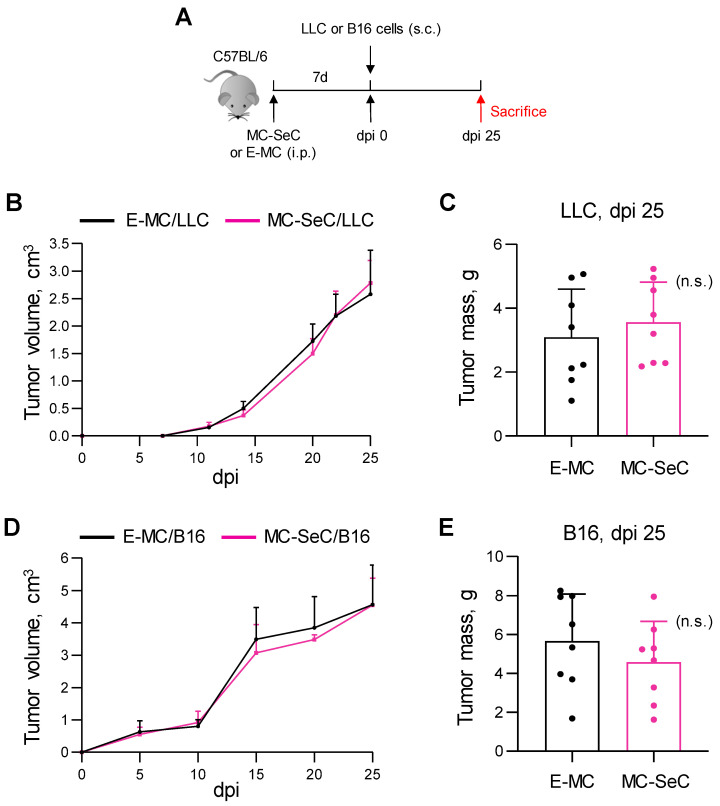
Grafted SeCs do not affect tumor growth in vivo. (**A**–**E**) C57Bl/6 mice were injected i.p. with MC-SeCs (1.0 × 10^6^ SeCs/g body weight) (n = 8) or equivalent amount of E-MCs (n = 8) prior to being injected s.c. with LLC or B16 cancer cells, and the animals were sacrificed after 25 days (**A**). The volumes of subcutaneous masses were measured over time as an index of tumor growth (**B**,**D**), and tumor masses were weighed after their excision at dpi 25 (**C**,**E**). n.s., not statistically significant.

**Figure 4 cells-13-00544-f004:**
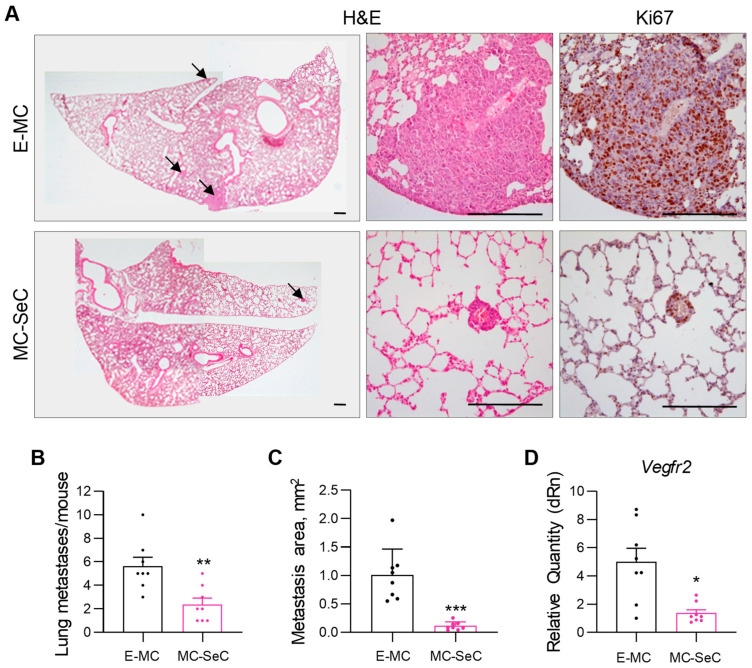
Grafted SeCs restrain lung metastasis formation. (**A**–**C**) Lungs isolated from LLC-bearing mice previously injected i.p. with MC-SeCs (n = 8) or E-MCs (n = 8) were analyzed for the presence of metastases. Reported are representative images of whole mounts of lungs (**A**, left panels; arrows indicate metastases) and higher magnification of representative metastases (**A**, middle panels) after hematoxylin–eosin staining. The immunostaining of consecutive slices for the proliferation marker Ki67 is shown (**A**, right panels). The numbers of lung metastases per mouse were evaluated (**B**). The mean areas of LLC metastasis in the lungs of E-MC- and MC-SeC-injected mice were determined (**C**). (**D**) The expression of the tyrosine kinase receptor gene *Vegfr2* in tumor masses of LLC-bearing E-MC- and MC-SeC-injected mice was evaluated using real-time PCR. Scale bars (**A**), 200 µm. * *p* < 0.05, ** *p* < 0.01, and *** *p* < 0.001.

## Data Availability

The original contributions presented in the study are included in the article and Appendix A; further inquiries can be directed to the corresponding author.

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
