# Peer review of "Grafted Sertoli Cells Exert Immunomodulatory Non-Immunosuppressive Effects in Preclinical Models of Infection and Cancer"

_cells, 2024, doi:10.3390/cells13060544_

Round 1
Reviewer 1 Report
Comments and Suggestions for Authors
This is a very nice study, demonstrating that despite their immunoregulatory properties, that allow for immune protection of germ cells and other tissues, Sertoli cells also provide antifungal resistance, do not alter tumor growth and decrease cancer metastasis. The paper is well written with excellent, compelling data. The results are very exciting and important. I have only a few minor comments.
1. Lines 65-67: There are papers indicating Sertoli cells express low levels of MHC class I and that IFNg can induce expression of MHC class II.
2. Line 230: I think this should be Figure 1A and 2A since it refers to both the lung and GI experiments.
3. Legend for figure 1: D and E are reversed in the figure legend. D should be about Fungal Conidia number and E should be BALF cell counts.
4. Lines 262 to 264: the text indicates significant changes in expression levels. However, not all of the genes are significantly changes. Maybe modify to indicate the significant changes. For the genes without significant changes, you could indicate the p values in the text.
Comments on the Quality of English LanguageN/A
Author Response
We wish to thank this Reviewer for the very positive evaluation of the manuscript and the useful comments provided, which have allowed us to improve the quality of the work. Following are our replies to the Reviewer’s comments.
- Lines 65-67: There are papers indicating Sertoli cells express low levels of MHC class I and that IFNg can induce expression of MHC class II.
Reply.
We thank the reviewer for the useful comment. We have rephrased the sentence and replaced the former Ref. 12. The reference order has been changed consequently.
- Line 230: I think this should be Figure 1A and 2A since it refers to both the lung and GI experiments.
Reply.
That is right; we have added the reference to Figure 2A in the text.
- Legend for Figure 1: D and E are reversed in the figure legend. D should be about Fungal Conidia number and E should be BALF cell counts.
Reply.
We apologize for the mistake. We have corrected the figure legend.
- Lines 262 to 264: the text indicates significant changes in expression levels. However, not all of the genes are significantly changes. Maybe modify to indicate the significant changes. For the genes without significant changes, you could indicate the p values in the text.
Reply.
That is right. The expression of Il1b was not statistically significant at dpi 3 and we were ready to remove it from the text. However, Il1b expression resulted statistically significant at dpi 7, which we have added based on the suggestion of one of the Reviewers. Thus, we have maintained the original sentence in the text referring it to the post-infection period (“…we found that the expression levels of IL-6 (Il6), IL-1β (Il1b) and IL-17A (Il17a) were significantly reduced by MC-SeC, whereas the expression of anti-inflammatory mediators, such as Ido1, Il10 and Il1rn were significantly increased during the post-infection period (Figure 1G).”). Moreover, the statistical significance has been added now to the graphs of Figure 1G where missing (i.e., Il17a and Il10); we apologize for that lack of clarity.
Reviewer 2 Report
Comments and Suggestions for Authors
Chiappalupi et al. have studied the effects of intraperitoneally grafted porcine microencapsulated Sertoli cells (MC-SeC) on the development of fungal infection (Aspergillus fumigatus in the lungs and Candida albicans in the colon) and cancer (Lewis lung carcinoma or B16 melanoma cells) using mice as a model system. The main results obtained suggest that MC-SeC primarily exhibit an immunoregulatory activity in both fungal infection and cancer. Also, MC-SeC appear to reduce Lewis lung carcinoma metastasis. These findings are important for further development of SeC-based therapeutic approaches.
Minor comments:
Figure 1H clearly shows the reduced expression of the Cyp1a1 gene in MC-SeC animals. Figure 2E shows that the expression level of this gene is increased at dpi 4 and decreased at dpi 10. However, the text in lines 283-285 ignores these facts. Accordingly, the statement/conclusion in lines 285-287 seems to be too strong.
The text in the line 347 (“grafting of SeC in tumor-bearing animals”) contradicts with the experimental setup (shown in Figure 3A and described in lines 108-111), in which SeC were injected before LLC/B16 cells.
The conclusion about the role of the Vegfr2 gene in lines 348-349 (“thus restraining cancer cell spread and organ colonization”) seems to be an overstatement, as no appropriate data provided.
RT-qPCR measurements for dpi 7 and dpi 10 are missed in Figures 1FG and 2CD, respectively.
The RT-qPCR results would be more solid if more than one reference gene is used. Also, what was the reason to use different reference genes for lungs/colons and tumor masses?
Consider changing “fold increase” to “fold change” in Figures with RT-qPCR results, as there is a decrease in mRNA level in some cases.
Lines 191-192 and 197, “Actb (β-actin)” and “Gapdh” should be written in italic, as the genes/mRNA are meant there.
In Figure 2A, dpi 0 is not indicated as “Sacrifice”, whereas some experimental data (RT-qPCR) for this timepoint is provided. How is it possible?
It is not clear, what exactly was used as a material for RNA isolation for the experiment shown in Figure 4D. The figure legend and the text (line 346) say slightly different things: “lung homogenates” and “LLC masses”. What is correct?
How were the “Relative Quantity” values shown in Figure 4D calculated?
The meaning of “dRn” should be explained.
In the line 360, “the tyrosine kinase receptor” should be change to “the tyrosine kinase receptor gene”.
Author Response
We wish to thank this Reviewer for her/his criticism in evaluating the manuscript and the useful comments and suggestions provided, which have allowed us to improve the quality of the work. Following are our replies to the Reviewer’s comments.
- Figure 1H clearly shows the reduced expression of the Cyp1a1 gene in MC-SeC animals. Figure 2E shows that the expression level of this gene is increased at dpi 4 and decreased at dpi 10. However, the text in lines 283-285 ignores these facts. Accordingly, the statement/conclusion in lines 285-287 seems to be too strong.
Reply
We thank the Reviewer for this comment. Actually, we found that MC-SeC but not E-MC promoted the expression of the AhR-dependent genes, Cyp1a1, Cyp1b1 and Il22 in colon fungal infection compared to controls (Figure 2E). Similarly, Cyp1b1 and Il22 resulted upregulated by MC-SeC in the lung infection model (Figure 1H). In this latter model, the expression of Cyp1a1 was found reduced in comparison with controls. This difference between the two experimental models may be due to the different organs and pathogens considered. We have rephrased the sentence and the concluding statement as follows, “We found that MC-SeC but not E-MC promoted the expression of these AhR-dependent genes in colon fungal infection compared to controls (Figure 2E). Similarly, Cyp1b1 and Il22 resulted upregulated by MC-SeC in the lung infection model (Figure 1H), suggesting an involvement of the AhR-dependent pathway also in this case. Altogether, these results pointed to a regulatory, non-immunosuppressive, activity of grafted MC-SeC in inflammation and tissue repair in infection.”.
- The text in the line 347 (“grafting of SeC in tumor-bearing animals”) contradicts with the experimental setup (shown in Figure 3A and described in lines 108-111), in which SeC were injected before LLC/B16 cells.
Reply
We apologize for the lack of clarity. Mice were always injected with MC-SeC before tumor cell injection. We have corrected the sentence.
- The conclusion about the role of the Vegfr2 gene in lines 348-349 (“thus restraining cancer cell spread and organ colonization”) seems to be an overstatement, as no appropriate data provided.
Reply
We agree with this Reviewer and we have rephrased the sentence as follows, “Thus, the presence of grafted SeC in LLC-bearing animals, while not affecting the primary cancer cell growth, restrained cancer cell spread and lung colonization, which may be linked to a reduced expression of Vegfr2 in primary tumors.”.
- RT-qPCR measurements for dpi 7 and dpi 10 are missed in Figures 1FG and 2CD, respectively.
Reply
We have added the missing time points in Figures 1F,G (dpi 7) and 2C,D (dpi 10).
- The RT-qPCR results would be more solid if more than one reference gene is used. Also, what was the reason to use different reference genes for lungs/colons and tumor masses?
Reply
Both Actb and Gapdh gene expression resulted not affected by treatments in our experimental conditions. The use of different reference genes was simply linked to the different experimental settings used by two independent laboratories involved in this work.
- Consider changing “fold increase” to “fold change” in Figures with RT-qPCR results, as there is a decrease in mRNA level in some cases.
Reply
Done.
- Lines 191-192 and 197, “Actb (β-actin)” and “Gapdh” should be written in italic, as the genes/mRNA are meant there.
Reply
Done.
- In Figure 2A, dpi 0 is not indicated as “Sacrifice”, whereas some experimental data (RT-qPCR) for this timepoint is provided. How is it possible?
Reply
We apologize for missing this time-point in the experimental scheme. We have modified the scheme of Figure 2A including dpi 0 as a sacrifice time-point.
- It is not clear, what exactly was used as a material for RNA isolation for the experiment shown in Figure 4D. The figure legend and the text (line 346) say slightly different things: “lung homogenates” and “LLC masses”. What is correct?
Reply
We apologize for the mistake. RNA isolation was from primary tumor masses in Figure 4D. We have corrected the figure legend.
- How were the “Relative Quantity” values shown in Figure 4D calculated? The meaning of “dRn” should be explained.
Reply
The Relative Quantity (DDCt method) is evaluated based on the “dRn” which is the baseline subtracted fluorescence reading normalized to the reference dye (ROX) as calculated by the Stratagene Mx3000P dedicated software. This has been added to the text of session 2.8. Real-time PCR.
- In the line 360, “the tyrosine kinase receptor” should be change to “the tyrosine kinase receptor gene”.
Reply
Done.